# Quantified EEG for the Characterization of Epileptic Seizures versus Periodic Activity in Critically Ill Patients

**DOI:** 10.3390/brainsci10030158

**Published:** 2020-03-10

**Authors:** Lorena Vega-Zelaya, Elena Martín Abad, Jesús Pastor

**Affiliations:** Clinical Neurophysiology; Hospital Universitario de La Princesa, C/Diego de León 62, 28006 Madrid, Spain; lorenacarolina.vega@salud.madrid.org (L.V.-Z.); emabad@salud.madrid.org (E.M.A.)

**Keywords:** Fast Fourier Transform, focal seizures, generalized seizures, periodic activity, rhythmic activity

## Abstract

Epileptic seizures (ES) are frequent in critically ill patients and their detection and treatment are mandatory. However, sometimes it is quite difficult to discriminate between ES and non-epileptic bursts of periodic activity (BPA). Our aim was to characterize ES and BPA by means of quantified electroencephalography (qEEG). Records containing either ES or BPA were visually identified and divided into 1 s windows that were 10% overlapped. Differential channels were grouped by frontal, parieto-occipital and temporal lobes. For every channel and window, the power spectrum was calculated and the area for delta (0–4 Hz), theta (4–8 Hz), alpha (8–13 Hz), and beta (13–30 Hz) bands and spectral entropy (Se) were computed. Mean values of percentage changes normalized to previous basal activity and standardized mean difference (SMD) for every lobe were computed. We have observed that BPA are characterized by a selective increment of delta activity and decrease in Se along the scalp. Focal seizures (FS) always propagated and were similar to generalized seizures (GS). In both cases, although delta and theta bands increased, the faster bands (alpha and beta) showed the highest increments (more than 4 times) without modifications in Se. We have defined the numerical features of ES and BPA, which can facilitate its clinical identification.

## 1. Introduction

Patients in intensive care units (ICU) usually have limited clinical neurological examination, either because of structural or functional altered conditions of the Central Nervous System (CNS) or due to the effects of drugs used for sedoanalgesia [1]. A very effective tool to evaluate the brain function in these conditions is electroencephalography (EEG). However, a dynamical evolution of injury is commonly observed in critically ill patients, due to the occurrence of epileptic seizures (ES), status epilepticus (SE), apoptosis, vasospasm or other different brain insults [2,3,4]. In this sense, continuous EEG (cEEG) is a non-invasive method that allows the functional assessment of the cerebral cortex in real time for prolonged periods of time. It has been proven to be an extraordinarily useful tool for detecting electrographic seizures and non-convulsive epileptic status (NCES), modifying treatment and assessing the functional prognosis [5,6,7,8,9].

In recent years, the development of mathematical analysis tools for bioelectric signals, commonly known by the acronym qEEG (quantified EEG), has introduced elements of objectivity into the analysis of EEG records [10]. In the ICU field, the qEEG has been applied to facilitate the interpretation of prolonged EEG recordings, as well as the identification of electrographic seizures [11,12,13,14].

The particularity of long-term electroencephalographic records of patients in ICU comprises the high frequency of artifacts that are found throughout the record and the rhythmic and periodic patterns frequently observed in these patients, which are easily confused with seizures. Both can be difficult to interpret not only on an EEG raw, but also with the qEEG tools currently used [15,16].

The time taken for a cEEG review is one of the most commonly given reasons for the use of qEEG in ICU and other diagnostic fields. However, we have taken a different approach to qEEG during cEEG: instead of just simplifying seizure detection, our aim is to obtain a comprehensive and efficient view of bioelectrical brain physiology in the most objective way [17]. To do this, we have developed a qEEG using classical mathematical methods, but in a neurophysiologically and clinically oriented fashion. We have used the assumption that EEG is founded in a homeostatic system [18,19] to obtain the main variables of our method, in order to establish an approximate direct relationship between the numerical magnitude variation and the underlying anatomo-functional system.

As stated above, the use of multiple drugs acting onto the CNS and the primary and secondary injuries profoundly affect the bioelectrical brain dynamics. Therefore, sometimes it is quite difficult to differentiate between bursts of periodic activity (BPA) and true ES/SE. However, this differentiation is of critical significance, affecting the functional or vital outcome of the patient to a high degree. The main problem is that EEG patterns analyzed de visu have not always the sharp morphology of ES/SE of non-ICU patients. Therefore, we need to consider a more physiopathological approach for an easy and effective identification.

The EEG is the multivariate spatio-temporal determination of the electrical potentials generated by the brain and recorded on the surface of the scalp. The oscillatory activity of the EEG, in clinical practice, is divided into four bands, depending on its oscillation frequency: delta (0–4 Hz), theta (4–8 Hz), alpha (8–13 Hz) and beta (13–30 Hz). Although the cerebral cortex is the main anatomical structure that generates these potentials [20], the regulation of the different EEG bands is carried out by different brain structures. This aspect is of extraordinary relevance, because its specific involvement in different pathologies will lead to specific changes in the different numerical variables obtained. See Hughes and John [21] for a detailed explanation of the model adopted here. In this model, beta activity is originated from cortico–cortical interactions and alpha, although generated at thalamus, has significant participation from cortical structures. This complex neuroanatomic homeostatic system is probably genetically determined and regulates baseline levels of local synchronization, global interactions between different regions and the spectral composition of the signal [22,23,24].

In this work, we propose a numerical method firmly based in the pathophysiology of CNS and evaluated whether the definition of a seizure as an abnormal increase in cortical activity can discriminate between ES and BPA, with quite similar morphological properties.

## 2. Materials and Methods

### 2.1. Patients and Definitions

We retrospectively reviewed the scalp EEG performed in patients admitted to the ICU as part of the clinical protocol. Indications included clinical suspicion of ES/NCSE or assessment for functional prognosis. Patients were over 18 years old and their medical history was reviewed. In all cases, the relatives of the patients gave free and informed consent to the procedures approved by the Hospital La Princesa Ethics Board.

According to the ILAE, an ES is a transient occurrence of signs and/or symptoms due to abnormal excessive or synchronous neuronal activity in the cortex of the brain [25]. Obviously, signs/symptoms are usually excluded in critically ill patients, but the excessive activity of the cortex is mandatory to a positive identification. ES were visually defined according to customary criteria [26]. On the other hand, we have defined BPA as all the transitory patterns including periodic discharges and rhythmic delta activity, according to the definition of the American Clinical Neurophysiology Society’s Standardized Critical Care EEG Terminology [27].

### 2.2. EEG Recording

EEGs’ records were performed using a 32-channel digital system (EEG32U, NeuroWorks, XLTEK^®^, Oakville, ON, Canada) with 19 electrodes placed according to a 10–20 international system. In addition, the differential derivation I of Einthoven for ECG was placed. If necessary, surface electromyography (EMG) channels were added. Recordings were performed at 512 Hz sampling rate, with a filter bandwidth of 0.5 to 70 Hz and notch-filter of 50 Hz. Electrode impedances were usually below 15 kΩ.

Clinical report was performed by a clinical neurophysiologist with several years of experience in ICU electroencephalography. qEEG was performed off-line; therefore, no information from this method was clinically relevant.

Artifact-free periods of either putative ES or BPA were selected and exported in ASCII file to be quantified (Quantification of EEG). All of these records included a basal period previous to the visual beginning of paroxysmal activity and a posterior time of at least 1 min each. The start and ending of significant activities were visually identified.

### 2.3. Quantification of EEG

The algorithm used was as follows:EEG channels were digitally filtered by a 6th order Butterworth digital filter between 0.5 and 30 Hz ECG channels were filtered at a bandwidth of 3–30 Hz. Notch filter (bandwidth 48–52 Hz) was also used [28];Differential EEG montage was reconstructed in a double-banana fashion. Topographical placement of differential channels was defined onto the scalp as the mid-point between the electrode pairs defining the channel, e.g., the Fp_1_–F_3_ channel would be placed at the mid-point of the geodesic between Fp1 and F3 electrodes;All the recording was divided in 1 s moving windows with 10% overlap. This windowing allowed us a frequency precision of 0.5 Hz. For each window (*n*) and frequency (*k*), we computed the Fast Fourier Transform (FFT) of the voltage ($V^{m}\left(n\right)$
) obtained from every differential channel (*m*) to obtain the power spectrum ($S_{n,k}^{m}$, in µV^2^/Hz). We used this expression:
(1)$$S_{n,k}^{m}=\overset{N-1}{\underset{n=0}{\sum }}V^{m}\left(n\right)e^{-i\frac{2\pi }{N}kn};m=Fp_{1},\;F_{3},\ldots$$
We computed also the Shannon spectral entropy (Se) according to
(2)$$Se_{k}^{m}=-\overset{F}{\underset{k=0}{\sum }}p_{k}\log_{2}p_{k}$$
where *F* is the maximum frequency computed and *p_k_* is the probability density of S, obtained from the expression
(3)$$p_{k}=\frac{S_{n,k}^{m}}{\sum_{k=0}^{F}S_{n,k}^{m}∆k}$$
We computed the area under the $S_{n,k}^{m}$ according to the classical segmentation of EEG bands. We used this expression:
(4)$$A_{j}\left(k\right)=\overset{sup}{\underset{k=inf}{\sum }}\;S_{n}^{m}\left(k\right)∆k;j=\delta ,\theta ,\alpha ,\beta$$


The expression *sup* refers to the upper limit of every EEG band. Areas of the same band were grouped by cerebral lobe. In the case of the left hemisphere (showed as example), we grouped in frontal $F=\left\{\frac{\left(Fp_{1}-F_{3}\right)+\left(F_{3}-C_{3}\right)+\left(Fp_{1}-F_{7}\right)}{3}\right\}$, parieto-occipital $PO=\left\{\frac{\left(C_{3}-P_{3}\right)+\left(P_{3}-O_{1}\right)+\left(T_{5}-O_{1}\right)}{3}\right\}$ and temporal $T=\left\{\frac{\left(Fp_{1}-F_{7}\right)+\left(F_{7}-T_{3}\right)+\left(T_{3}-T_{5}\right)+\left(T_{5}-O_{1}\right)}{4}\right\}$. Channels from right hemisphere were accordingly grouped. These areas, for both bands (*j*) and lobes (*r*), $A_{j}^{r}\left(t\right);\;r=F,\;PO,\;T$, were plotted as time functions and compared between both hemispheres. The same groups were used to compute Se.

Numerical analysis of EEG recordings was performed with custom-made Matlab^®^ R2019 software (MathWorks, Natic, MA, USA) [17].

### 2.4. Statistics

Statistical comparisons between groups were performed using the *z*-score, the Student’s *t*-test or ANOVA for data with normal distributions. Normality was evaluated using the Kolmogorov–Smirnov test. The Mann–Whitney Rank sum test or ANOVA on ranks were used when normality failed. In the last case, the Tukey test was used for all pairwise comparison of the mean ranks of the treatment groups. SigmaStat^®^ 3.5 software (SigmaStat, Point Richmond, CA, USA) was employed for statistical analysis.

We have used two different measures to evaluate changes:Difference between normalized data (*norm_band_:band =* δ, θ, α, β, Se). For every patient and every variable (δ, θ, α, β, Se), we computed the mean value during basal, burst (ES or BPA) and post-burst states. All the variables were normalized to basal state, considered as 100%;Standardized mean difference (SMD). With the aim to use a common metric to evaluate changes in different kinds of measures, we used the equation [29]:
(5)$$\mathrm{SMD}=\frac{\mathrm{previous}\;\mathrm{measure}-\mathrm{posterior}\;\mathrm{measure}}{\mathrm{pooled}\;\mathrm{standard}\;\mathrm{deviation}}$$



This index is most apt to evaluate size effects when comparing changes in different types of measures. To calculate the SMD, measures prior to the burst were computed. Measures during burst and post-burst were calculated, too. Since positive SMD imply decreased values of the evaluated measure, SMD values were multiplied by −1 for a more intuitive visualization.

Although both methods use the difference between a basal measurement and a posterior one, they are clearly different, because in the first case, the difference is performed between normalized values and in the last one, between true measured values. We have used both in order to uncover spurious effects due to normalization.

In case of focal ES, we pooled results by the epileptic hemisphere, either right or left, defined as the hemisphere where the ictal pattern starts.

The significance level was set at *p* = 0.05. Results are shown as the mean ± SEM, except where otherwise indicated. The inter-percentile 25–75 range is indicated between brackets and median is refereed as *Med.*

## 3. Results

We have analyzed 18 BPA from 15 patients. Clinical data are shown in Table 1. When BPA were taken from the same patient, we used recordings from different days. We also analyzed 21 recordings of ES, obtained from different patients, whose clinical features are shown in Table 2.

No significant differences in timing of appearing BPA/ES have been observed. However, a more systematic approach should be implemented to address this aspect.

ES were classified as generalized seizures (GS) in six cases, while in the rest, seizures were focal (FS) (ILAE 1981). Although rhythmic and periodic patterns can be either generalized or lateralized; only the BPA of the first group has been included in this work.

In Figure 1, a typical ES and BPA are visually identified (see also Figure A1 and Figure A2, Appendix A).

### 3.1. Features of BPA

We have computed the normalized changes in mean power square of all the band and entropy for every lobe during BPA (Figure 2). Only delta band significantly increased during BPA, but it does near 3-times in frontal lobes ($norm_{\delta }^{left}=298.2\pm 22.5;\;Med=247.8;\left[155.7\text{–}457.9\right]$; *p* < 0.001 ANOVA on ranks; $SMD_{\delta }^{left}=0.848\pm 0.103;\;Med=0.988;\left[0.682\text{–}1.155\right]$; *p* < 0.001 ANOVA on ranks; $norm_{\delta }^{right}=293.9\pm 64.2;\;Med=208.0;\left[179.0\text{–}316.0\right]$; *p* < 0.001 ANOVA on ranks; $SMD_{\delta }^{rigth}=0.912\pm 0.093;\;Med=0.965;\left[0.879\text{–}1.079\right]$; *p* < 0.001 ANOVA on ranks), the double in parieto-occipital lobes ($norm_{\delta }^{left}=215.6\pm 24.1;\;Med=177.6;\left[134.8\text{–}291.8\right]$; *p* < 0.001 ANOVA on ranks; $SMD_{\delta }^{left}=0.743\pm 0.091;\;Med=0.733;\left[0.409\text{–}1.042\right]$; *p* < 0.01 ANOVA on ranks; $norm_{\delta }^{right}=217.5\pm 24,2;\;Med=197.3;\left[145.2\text{–}241.3\right]$; *p* < 0.001 ANOVA on ranks; $SMD_{\delta }^{rigth}=0.749\pm 0.112;\;Med=0.870;\left[0.539\text{–}1.107\right]$; *p* < 0.01 ANOVA on ranks) and 3-times in temporal lobes ($norm_{\delta }^{left}=325.9\pm 66.4;\;Med=207.1;\left[135.3\text{–}398.3\right]$; *p* < 0.001 ANOVA on ranks; $SMD_{\delta }^{left}=0.748\pm 0.115;\;Med=0.886;\left[0.465\text{–}1.131\right]$; *p* < 0.05 ANOVA on ranks; $norm_{\delta }^{right}=267.7\pm 39.4;\;Med=262.0;\left[144.4\text{–}301.0\right]$; *p* < 0.001 ANOVA on ranks; $SMD_{\delta }^{right}=0.863\pm 0.114;\;Med=0.953;\left[0.612\text{–}1.231\right]$; *p* < 0.001 ANOVA on ranks). The rest of bands did not significantly change during BPA.

No differences between right and left lobes were observed for any band (paired Student *t*-test).

Spectral entropy significantly decreased between 10% and 20% during BPA in frontal ($norm_{\delta }^{left}=88.1\pm 2.82;\;Med=84.9;\left[79.4\text{–}98.7\right]$; *p* < 0.001 ANOVA on ranks; $SMD_{\delta }^{left}=-848\pm 0.103;\;Med=-0.988;\left[-0.659\text{–}-1.159\right]$; *p* < 0.01 ANOVA on ranks; $norm_{\delta }^{rigth}=88.1\pm 2.4;\;Med=87.8;\left[79.8\text{–}94.9\right]$; *p* < 0.001 ANOVA on ranks; $SMD_{\delta }^{right}=-0.912\pm 0.093;\;Med=-0.965;\left[-0.877\text{–}-1.083\right]$; *p* < 0.05 ANOVA on ranks), parieto-occipital ($norm_{\delta }^{left}=91.8\pm 2.0;\;Med=90.2;\left[84.6\text{–}96.9\right]$; *p* < 0.01 ANOVA on ranks; $SMD_{\delta }^{left}=-0.734\pm 0.091;\;Med=-0.733;\left[-0.386\text{–}-1.047\right]$; *p* < 0.05 ANOVA on ranks; $norm_{\delta }^{right}=90.5\pm 2.3;\;Med=89.6;\left[84.4\text{–}97.5\right]$; *p* < 0.01 ANOVA on ranks; $SMD_{\delta }^{rigth}=-0.749\pm 0.112;\;Med=-0.870;\left[-0.520\text{–}-1.107\right]$; *p* < 0.05 ANOVA on ranks) and temporal lobes ($norm_{\delta }^{left}=89.7\pm 2.8;\;Med=90.5;\left[82.4\text{–}97.6\right]$; *p* < 0.001 ANOVA on ranks; $SMD_{\delta }^{left}=-0.748\pm 0.115;\;Med=-0.886;\left[-0.394\text{–}-1.143\right]$; *p* < 0.01 ANOVA on ranks;$norm_{\delta }^{right}=90.2\pm 2.17;\;Med=89.1;\left[82.2\text{–}97.4\right]$; *p* < 0.01 ANOVA on ranks; $SMD_{\delta }^{right}=-0.863\pm 0.114;\;Med=-0.953;\left[-0.561\text{–}-1.247\right]$; *p* < 0.01 ANOVA on ranks).

No differences between lobes were observed for Se (paired Student *t*-test).

### 3.2. Features of Generalized Seizures

In the case of GS, the pattern induced in the power spectral was completely different from BPA, as we can observe from Figure 3. There was a generalized increase in all the bands without differences between the different lobes, increasing in power from the slower (δ and θ) to the faster (α and β) bands. Delta bands increased around 1.5 times (Table A1, Appendix A), θ and α bands increased approximately to double (Table A2 and Table A3, Appendix A) the mean, while β bands increased between 3 and 4 times (Table A4, Appendix A), except at the right temporal lobe. However, *Se* did not change in GS (Table A5, Appendix A).

No differences between the right and left lobes were observed (paired Student *t*-test) for any band. Therefore, a true global and symmetrical participation for all the scalp was observed in GS.

### 3.3. Features of Focal Seizures

The pattern induced in the case of FS in spectral powers was also completely different from BPA and quite similar to GS, as we can observe from Figure 4. It must be realized that FS started in a well-defined region, but what we have computed was the overall activity during the previous ictal and post-ictal periods. We have observed a generalized increase in all the bands, increasing in power from the slower (δ and θ) to the faster (α and β) bands. Delta bands increased by around 1.5 times (Table A6, Appendix B), θ and α bands increased by approximately double (Table A7 and Table A8, Appendix B) and β bands augmented between 3–4 times (Table A9, Appendix B), except for the right temporal lobe. As we observed in GS, *Se* did not change in FS (Table A10, Appendix B).

No differences between right and left lobes were observed (paired Student *t*-test) for any band. Therefore, a true global and symmetrical participation for all the scalp was observed, although we have defined the origin of these seizures as focal.

### 3.4. Comparison of Focal and Generalized Seizures

We have compared all the electroencephalographic bands by lobes in GS and FS. We have not found differences in δ, θ and α bands, or in *Se*. We only have found differences for β band and, even in this case, in a non-significant way. In fact, we have observed an excess of normalized beta for GS at the right frontal (*p* = 0.022, Mann–Whitney, not significant for SMD − *p* = 0.056) and parieto-occipital lobes (*p* = 0.032, Mann–Whitney, not significant for SMD − *p* = 0.105) and a decrease at the ipsilateral temporal lobe (*p* = 0.011, Mann–Whitney − *p* = 0.043 for SMD). These results show a high inconsistency, and, therefore, it is not possible to consider the spectral structure of GS and FS to be different.

It is relevant to observe that FS always evolved to GS, affecting the entire cortex with a similar pattern.

### 3.5. Numerical Definition of ES

Taking into account that GS and FS were similar, we pooled together and compared them with BPA (Figure 5). We have used data from right and left hemispheres, considering that there are no differences between them. Delta power for the frontal lobe was higher for BPA than for ES, but for the rest of the bands, the power was higher in the case of ES.

The difference between both states allowed us to define numerical criteria to separate ES from BPA. We consider that a paroxysmal event in a critically ill patient would be an ES with a high probability when the values for normalized increments of bands at different places of the scalp are included into the range of values defined by IP25–75 for ES and outside of the range of IP25–75 for BPA superimposed to ES. In Table 3, we show the IP25–75 for ES and BPA at the different lobes and the superposition between both intervals for the same band and lobe, expressed as a percentage.

From Table 3, we can observe that superposition is very high for δ band (i.e., this band is not discriminative), low for θ and Se and practically null for α and β bands. Therefore, the intervals for increments of normalized activity defining an ES in these types of patients are as follows (excluding superposition and rounding); $\delta_{F}=\left[119,166\right]$; $\theta_{F}=\left[173,264\right]$; $\theta_{PO}=\left[168,248\right]$; $\theta_{T}=\left[151,274\right]$; $\alpha_{F}=\left[159,244\right]$; $\alpha_{PO}=\left[159,244\right]$; $\alpha_{T}=\left[159,244\right]$; $\beta_{F}=\left[141,374\right]$; $\beta_{PO}=\left[146,262\right]$; $\beta_{T}=\left[141,374\right]$; $Se_{F}=\left[97,110\right]$; $Se_{PO}=\left[98,107\right]$; $Se_{T}=\left[98,104\right]$.

## 4. Discussion

In this work, we have defined in a numerical way the features for BPA and ES in critically ill patients. In other words, we have identified the differences between both paroxysmal states. This finding is very important from a clinical point of view, because, in a high degree of patients, the morphological features of recordings are equivocal and it is extremely difficult to differentiate between seizures and non-epileptic bursts. We have used the pathophysiological definition of epilepsy and we have observed that ES always implies an increase in cortical bands (α and β), while in BPA. these bands did not change.

The qEEG analyses currently used are displayed as hemispheric averages. The most frequently used tools are compressed spectral array (CSA), data based on amplitude (amplitude-integrated EEG; envelope trend), rhythmicity (rhythmicity spectrogram), or spectral symmetry (asymmetry index and spectrogram) [13,30]. The effectiveness of these tools in identifying seizures has been reported in other studies, either using only one of them [14,16,30] or analyzing a panel of multiple qEEG trends analyzed by experts [13]. Although the sensitivity cannot be considered negligible, a very low efficiency has been reported when identifying seizures with a low voltage, short duration, seizures that do not extend at least through a hemisphere, or that occurred in the context of abundant interictal epileptiform discharges. A comparison of our method with those previously described was out of the scope of this work.

ES in critically ill patients with encephalopathy show slower frequencies, longer duration and have less clearly defined onset, evolution and offset than seizures in awake patients, resulting in more difficulty to identify, especially because BPA are quite similar in morphology and usually frequent in these kinds of patients [31]. Our approach to differentiation between BPA and ES has been found in the pathophysiological definition of epilepsy [25]. We hypothesized that ES should necessarily show an increase of cortical activity. Beta band is exclusively originated at cortex and alpha band although thalamic in origin, has a huge cortical participation, with ten times more afferences from cortex to thalamus than thalamic efferences [32,33]. Therefore, alpha and beta bands must necessarily increase in ES and must be considered as some kind of landmark for this condition. On the contrary, BPA should not increase cortical bands (by definition), at least, at the same degree than during ES. Delta activity is assumed to originate in oscillator neurons in deep cortical layers and in the thalamus and probably reflects hyperpolarization of cortical neurons [21,23]. Although delta and theta bands also imply cortical synapses, the frequency resonances are completely different from alpha and beta and its participation in epilepsy is lower. From several years ago, we know in clinical practice that delta and theta are more involved in encephalopathy than faster bands as alpha and beta [31]. These hypotheses have been well reinforced by the results obtained in this work.

In the present work, we have analyzed just ES affecting the entire scalp. In fact, we have identified more FS than GS, but, in all cases, seizures spread. Considering that we have computed periods of several seconds (even of dozens of seconds), it is not surprising that FS and GS share similar spectral properties.

Nonetheless, we propose that the method used, based on the pathophysiology of epilepsy, can be easily generalized to true focal seizures, because for every cortical region, activity must be increased, therefore, increasing alpha and beta bands. It is only necessary to apply the numerical definition to a single hemisphere, or even to a single lobe. The same can be said for hemispheric periodic discharges, because the definition is based on physiological concepts.

Although we have defined the method for critically ill patients, we can extend its application to other patients. Generally, ES are easy to identify but, in some patients, (e.g., patients with severe cognitive impairments, cerebral injuries and atypical clinical manifestations) it is difficult to differentiate behavioral paroxysmal events from true ES. According to the definition of epilepsy, even in these cases, cortical activity must be increased. Therefore, we can use this method to exclude epilepsy in those cases where α/β activity do not change (or even decrease) during the event.

The method described can be implemented to automatically detect paroxysmal events during long-term monitoring of ICU patients.

Although the features described for BPA and ES in this work are robust and apparently well-defined, obviously we need to increase the number of patients in order to obtain a sharper definition of both states. Additionally, we need to check our hypothesis that the method can also discriminate for hemispheric or lobar paroxysmal activities.

## 5. Conclusions

We have defined the numerical features of ES and BPA in critically ill patients using the pathophysiological definition of epilepsy. This will facilitate its identification in clinical practice, allowing a precocious and more adequate treatment. 

## 6. Patents

The numerical method used in this work is being evaluated for patent Multivariate analysis method in EEG. Application number: P201930036; Application date: 01/21/2019. Pastor, J; Vega-Zelaya, L.

## Figures and Tables

**Figure 1 brainsci-10-00158-f001:**
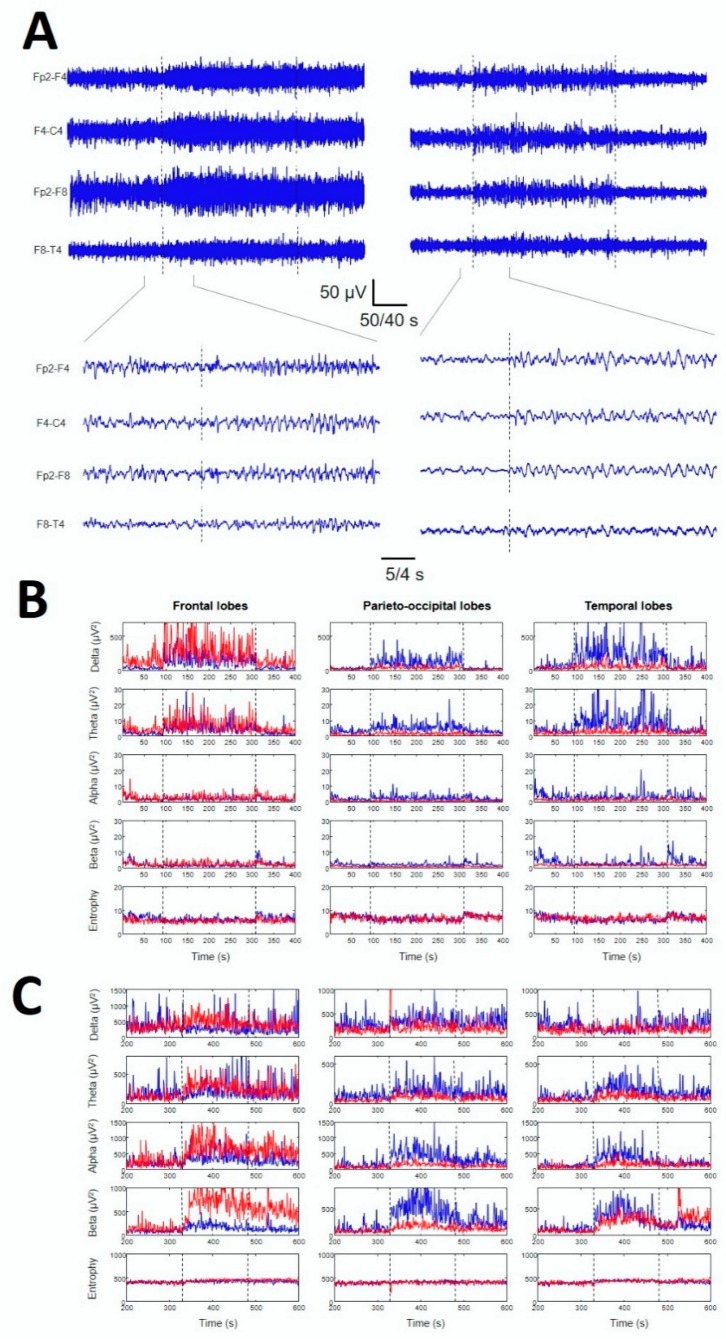
Examples of paroxysmal activity recorded in critically ill patients. (**A**) Left column: recordings from the right frontal lobe of a patient suffering a generalized seizure (GS). Right column: recordings from the same channels of a patient suffering a BPA. Traces are expanded below. Time-base is different for every recording. (**B**) Dynamics of bands (rows) and Se (lower row) for different lobes (columns) from the BPA above. (**C**) Dynamics of bands (rows) and Se (lower row) for different lobes (columns) from the GS above. Paroxysmal activity is indicated by vertical dotted lines. Red: right hemisphere. Blue: left hemisphere.

**Figure 2 brainsci-10-00158-f002:**
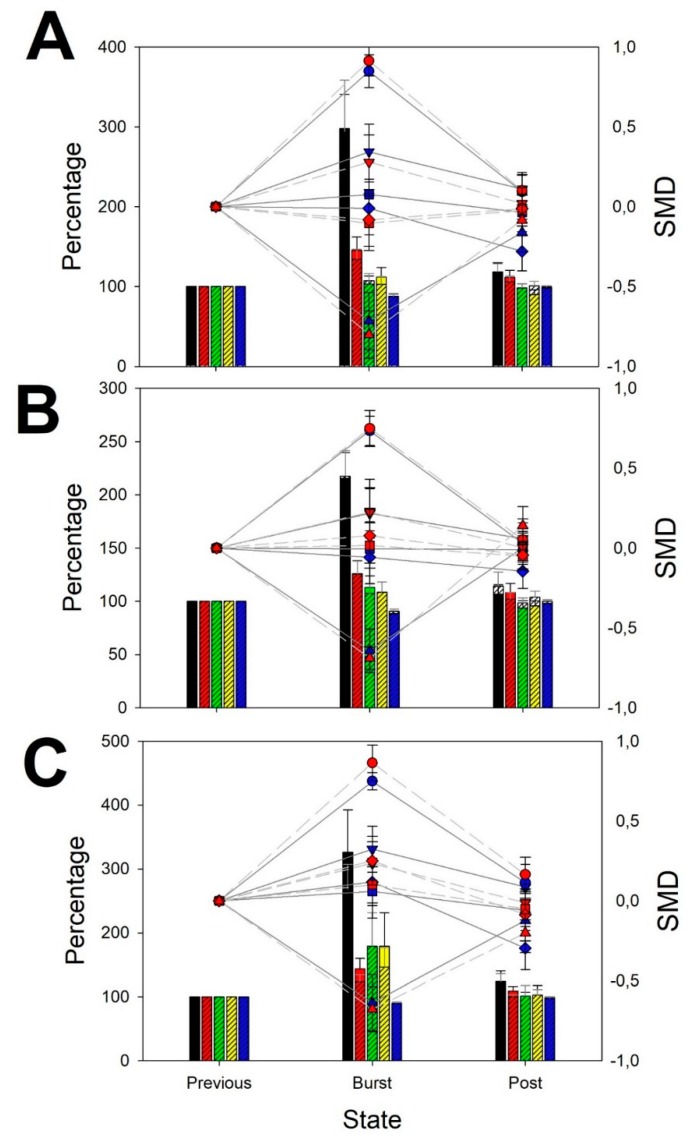
Normalized changes in power spectra and standardized mean difference (SMD) during BPA. (**A**) Frontal lobes; (**B**) Parieto-occipital lobes and (**C**) Temporal lobes. Bar graph (left *y*-axis): for every band and entropy, data from left (not-shaded bars) and right hemisphere (shaded bars) are superimposed (δ = black, θ = red, α = green, β = yellow, Se = blue). Dot graph (right *y*-axis): SMD for every left (blue) and right hemisphere (red) for every band and entropy (δ = circle, θ = down triangle, α = square, β = diamond, Se = up triangle).

**Figure 3 brainsci-10-00158-f003:**
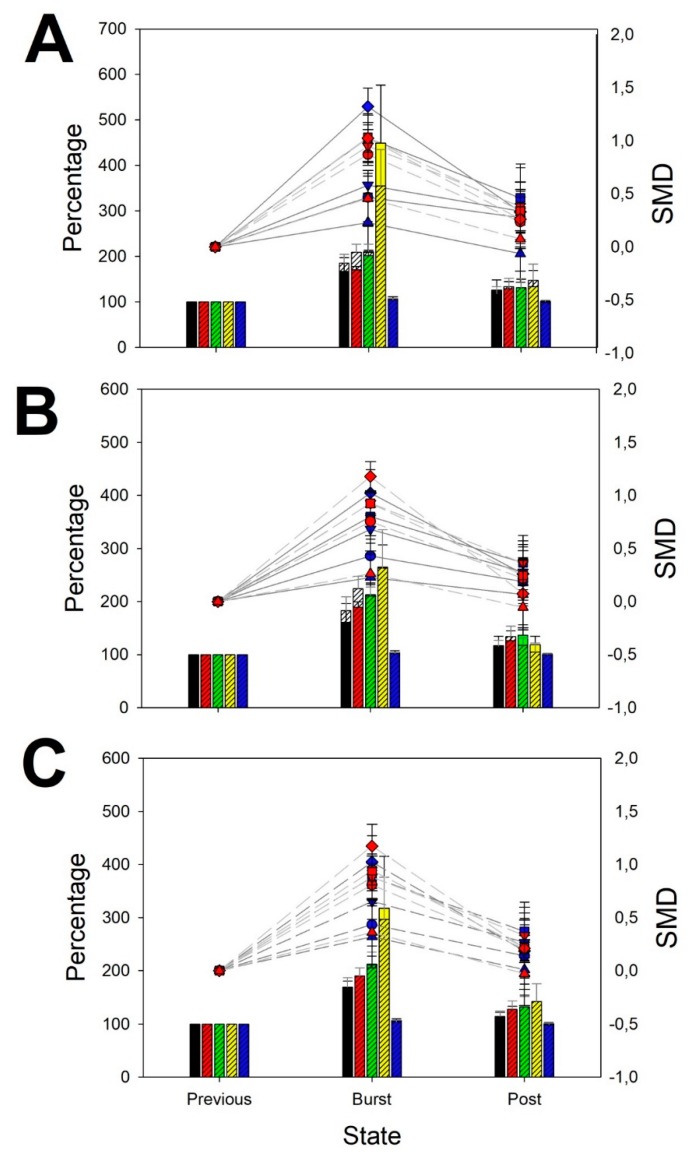
Normalized changes in power spectra and SMD during GS. (**A**) Frontal lobes; (**B**) parieto-occipital lobes and (**C**) temporal lobes. Bar graph (left *y*-axis): for every band and entropy, data from left (not-shaded color) and right hemisphere (shaded color) are superimposed (δ = black, θ = red, α = green, β = yellow, Se = blue). Dot graph (right *y*-axis): SMD for every left (blue) and right hemisphere (red) for every band and entropy (δ = circle, θ = down triangle, α = square, β = diamond, Se = up triangle).

**Figure 4 brainsci-10-00158-f004:**
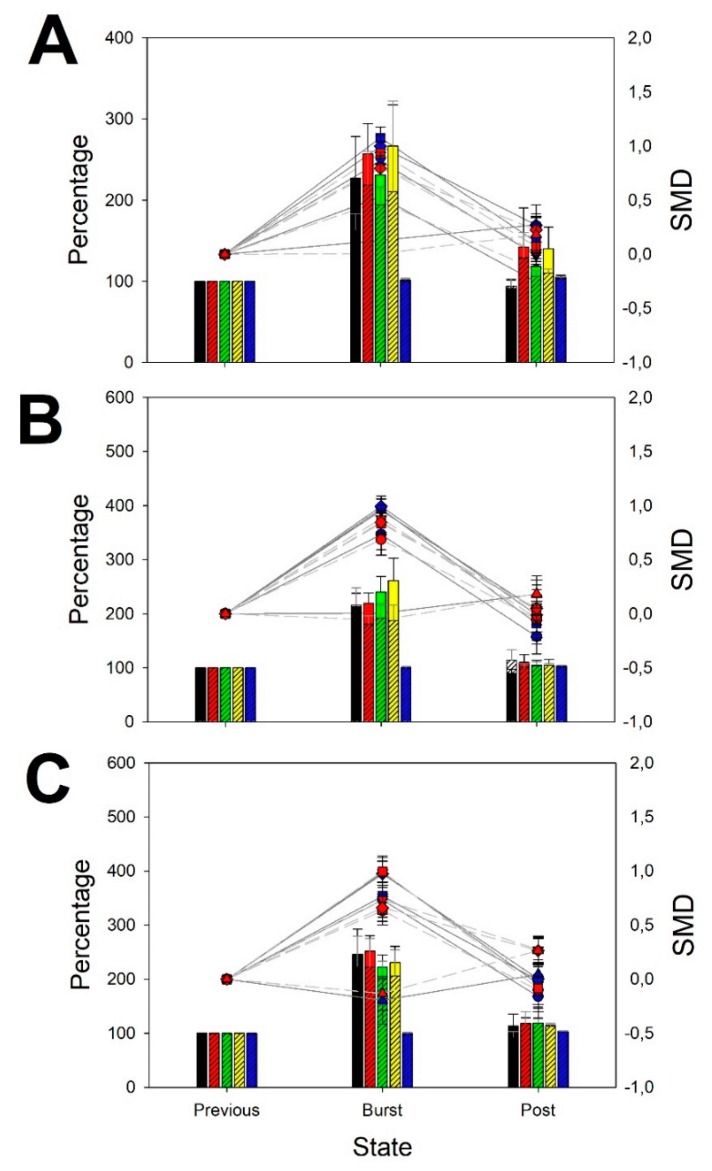
Normalized changes in power spectra and SMD during FS. (**A**) Frontal lobes; (**B**) parieto-occipital lobes and (**C**) temporal lobes. Bar graph (left *y*-axis): for every band and entropy, data from left (not-shaded color) and right hemisphere (shaded color) are superimposed (δ = black, θ = red, α = green, β = yellow, Se = blue). Dot graph (right *y*-axis): SMD for every left (blue) and right hemisphere (red) for every band and entropy (δ = circle, θ = down triangle, α = square, β = diamond, Se = up triangle).

**Figure 5 brainsci-10-00158-f005:**
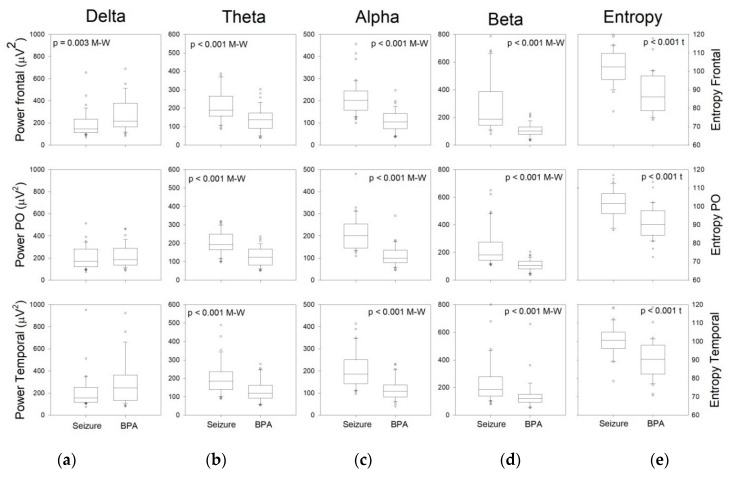
Comparison by bands and lobes between ES and BPA. (**a**) Delta; (**b**) Theta; (**c**) Alpha; (**d**) Beta; (**e**) Entropy. Probability is indicated when difference between ES and BPA is significant. M–W = Mann–Whitney rank sum test, t = Student *t*-test, PO = parieto-occipital.

**Table 1 brainsci-10-00158-t001:** Clinical features of patients showing bursts of periodic activity (BPA).

Patient	Age	Gender	Clinical Diagnosis	EEG Indication	EEG Findings	Sedation	AEDs
1	67	M	IH	AEDM	EA	Propofol	PHT/LVT
2	67	M	IH	AEDM	Cortical disturbance	Propofol	PHT/VPA
3	56	F	IH	Suspicion ES	EA	No	LVT
4	53	F	SAH	AEDM	EA	No	LVT
5	45	F	IH	Suspicion ES	Encephalopathy	No	VPA/LVT
6	45	F	IH	SDM	Encephalopathy	Propofol	VPA/LVT
7	58	F	RCRA	Abnormal movements	EA	No	PHT/LVT
8	58	F	RCRA	AEDM	EA	No	PHT
9	53	F	SAH	SDM	EA	Propofol	LVT
10	68	M	TBI	AEDM	Encephalopathy	No	VPA
11	68	M	TBI	AEDM	Encephalopathy	No	-
12	53	M	Meningitis	Impaired consciousness	EA	No	LVT
13	65	M	SAH	AEDM	EA	No	PHT/LVT
14	65	M	SAH	AEDM	Encephalopathy	No	PHT/LVT
15	42	F	Metabolic disorder	AEDM	Encephalopathy	No	PHT/LVT

EEG: electroencephalography; AEDs: antiepileptic drugs; M: male; F: female; IH: Intracerebral hemorrhage; SAH: Subarachnoid hemorrhage; RCRA: recovered cardiorespiratory arrest; TBI: traumatic brain injury; AEDM: antiepileptic drug monitoring; ES: epileptic seizures; SDM: sedation descent monitoring; EA: epileptiform activity; PHT: phenytoin; LVT: Levetiracetam; VPA: Valproic Acid; LVT: Levetiracetam.

**Table 2 brainsci-10-00158-t002:** Clinical features of patients showing seizures.

Patient	Age	Gender	Clinical Diagnosis	EEG Indication	EEG Findings	Sedation	AEDs
1	66	M	GCSE	SDM	Focal NCES	Propofol	PHT/LVT/LCM
2	63	F	Cerebral infarction	Abnormal movements	Focal NCES	Propofol	LVT
3	67	F	Postoperative CVS	Unexplained coma	Focal NCES	No	LVT
4	74	F	Postoperative CVS	Suspicion NCES	Focal NCES	Propofol	PHT/LVT
5	74	F	Postoperative CVS	Suspicion NCES	Focal NCES	Propofol	PHT/LVT
6	66	F	SAH	Unexplained coma	Focal NCES	No	LVT
7	64	F	Cerebral infarction	Suspicion NCES	Focal NCES	No	PHT/LVT/LMT
8	49	M	AVM	Suspicion ES	Focal NCES	No	-
9	57	F	Neurological deficit *****	Suspicion ES	Focal NCES	No	LVT
10	45	F	IH	Unexplained coma	Focal NCES	Propofol/MDZ	VPA/LVT
11	79	F	Viral encephalitis	Suspicion ES	Focal NCES	No	LVT/LMT
12	79	F	Viral encephalitis	SDM	Focal NCES	Propofol	LVT/LMT
13	79	F	Viral encephalitis	SDM	Focal NCES	Propofol	LVT/LMT
14	67	F	Refractory GCSE	SDM	Focal NCES	Propofol	VPA/LVT/CLZ
15	67	F	Refractory GCSE	SDM	Focal NCES	Propofol	VPA/LVT/CLZ
16	86	F	TBI	Unexplained coma	GSE	No	PHT/LVT/LCM
17	42	F	Metabolic disorder	Impaired consciousness	GNCSE	No	LVT
18	71	M	Postoperative CVS	Abnormal movements	GNCSE	MDZ	LVT
19	67	F	IH	SDM	GNCSE	Propofol/MDZ	LVT
20	53	M	Meningitis	Impaired consciousness	GNCSE	No	VPA/LVT
21	53	M	Meningitis	Impaired consciousness	GNCSE	Propofol/MDZ	VPA/LVT

*: discarded a vascular origin. GCSE: generalized convulsive status epileptic; CVS: cardiovascular surgery; SAH: Subarachnoid hemorrhage; AVM: arteriovenous malformation; IH: Intracerebral hemorrhage; TBI: traumatic brain injury; SDM: sedation descent monitoring; NCES: non-convulsive epileptic seizures; ES: epileptic seizures; GSE: generalized seizure epileptic; GNCSE: generalized non-convulsive seizure epileptic; MDZ: midazolam; PHT: phenytoin; LVT: Levetiracetam; LCM: Lacosamide; LMT: Lamotrigine; VPA: Valproic Acid; CLZ: Clonazepam.

**Table 3 brainsci-10-00158-t003:** Inter-percentile 25–75 intervals for bands and lobes in BPA and ES. Superposition is indicated with respect to ES interval.

Variable	State	Frontal	Sup (%)	Parieto–Occipital	Sup (%)	Temporal	Sup (%)
Delta	ES	111.8–234.0	55.4	123.8–276.8	91.3	118.4–247.6	86.1
	BPA	166.3–370.4		137.1–278.7		136.3–253.9	
Theta	ES	158.2–264.0	14.2	166.7–247.7	1.6	139.2–230.5	19.9
	BPA	92.8–173.2		84.3–168.0		93.1–157.4	
Alpha	ES	158.8–244.0	0	146.0–248.6	0	144.3–243.5	0
	BPA	75.0–137.8		79.7–134.1		82.7–135.6	
Beta	ES	141.9–373.6	0	146.7–261.8	0	136.5–274.4	10.4
	BPA	77.1–137.8		82.2–137–2		95.1–150.8	
Entropy	ES	95.5–109.5	10.7	96.2–106.7	12.4	96.1–104.8	19.5
	BPA	78.9–97.0		84.3–97.5		82.3–97.8	

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
