# Peer review of "Quantified EEG for the Characterization of Epileptic Seizures versus Periodic Activity in Critically Ill Patients"

_brainsci, 2020, doi:10.3390/brainsci10030158_

Round 1
Reviewer 1 Report
Although the aim of the article by Zelaya et al. is stated clearly, the number of BPA and ES could be implemented. Moreover, it should be discussed the reason (and the possible meaning) why some bands increase/decrease in specific conditions but not in others. The discussion should be improved basing on the literature.
In addition to this:
Lines 265-266: “The only results…in a non-convulsive way.” This phrase is not clear.
In Figure 5, the meaning of the abbreviation PO should be detailed.
Author Response
We acknowledge the reviewer by the incisive questions raised and the goal in clarifying the manuscript. He/she has done a clever and detailed analysis of the manuscript.
Although the aim of the article by Zelaya et al. is stated clearly, the number of BPA and ES could be implemented. We have understood the word implemented by increased. We are conscious that the number of patients and ES/BPA is relatively scarce, therefore, we stated at discussion “obviously we need to increase the number of patients in order to obtain a sharper definition of both states” (lines 375-376). However, the differences between both groups are high enough to demonstrate our goal, i.e objectively identify numerical features for both groups. Therefore, although conscious that always a greater number of patients should be better, we think that the number analyzed is not an obstacle to accept the conclusions of our work.
Moreover, it should be discussed the reason (and the possible meaning) why some bands increase/decrease in specific conditions but not in others. We have added a paragraph (lines 341-356) clarifying this cleaver observation.
The discussion should be improved basing on the literature. We have added more cites.
In addition to this:
Lines 265-266: “The only results…in a non-convulsive way.” This phrase is not clear. We have modified the sentence by: We only have found differences for β band and, even in this case, in a non-significant way (lines 286-287).
In Figure 5, the meaning of the abbreviation PO should be detailed. We have added the meaning at legend (line 305).
Reviewer 2 Report
This is an interesting paper that is relevant to electroencephalographers and clinicians trying to differentiate these two entities, which can be challenging in the ICU setting.
- Just to clarify, the patients were divided into groups (ES, PBA) based on the interpretation of the EEGs by the original neurophysiologist’s interpretation and then records were quantified?
How difficult were these records for the original neurophysiologist to be certain of ES vs BPA? The point is that making this distinction is difficult.
- Was there a difference in timing of BPA vs ES? For example, was ES noted earlier in the recording?
- For Figure 1, an EEG example of focal seizure would also useful.
- For table 1 and 2, under sedation, were those only meds used? Were other AED’s used, especially if seizures were suspected?
- Explain AEMD: anti-epileptic drug monitoring as indication for EEG. This seems more prevalent in the BPA group that ES group.
- Did those with seizures have any clinical relevant features, different from BPA (ie.Changes in vitals, subtle motor movements, etc.) Were these features used in the original neurophysiologist’s interpretation?
- Did any patients have both findings? Often, these findings can be seen together.
Author Response
We acknowledge the reviewer by the adroit and incisive questions raised and the goal in clarifying the manuscript. He/she has done a clever and detailed analysis of the manuscript.
This is an interesting paper that is relevant to electroencephalographers and clinicians trying to differentiate these two entities, which can be challenging in the ICU setting.
1. Just to clarify, the patients were divided into groups (ES, PBA) based on the interpretation of the EEGs by the original neurophysiologist’s interpretation and then records were quantified? How difficult were these records for the original neurophysiologist to be certain of ES vs BPA? The point is that making this distinction is difficult. The reviewer is right about this point. Sometimes the identification was quite difficult and in some patients, we discussed nearly one hour before to emit the report. However, as stated in Material and Methods/EEG recording, no information from qEEG was used to identify ES/PBA. We have removed the word immediately at line 104 to stress this point.
2. Was there a difference in timing of BPA vs ES? For example, was ES noted earlier in the recording?. Non-significant results were obtained in this sense. We have added a sentence at lines 191-192.
3. For Figure 1, an EEG example of focal seizure would also useful. To avoid overcharge the figure 1, we have added an example of FS at Appendix A, Figure 2.
4. For table 1 and 2, under sedation, were those only meds used? Were other AED’s used, especially if seizures were suspected?. Of course, we have added a column to both tables with AEDs.
5. Explain AEMD: anti-epileptic drug monitoring as indication for EEG. This seems more prevalent in the BPA group that ES group. The reviewer is absolutely right. However, the column EEG indication reflects the perception of ICU physicians to solicit the first EEG and not the final diagnosis after EEG monitoring.
6. Did those with seizures have any clinical relevant features, different from BPA (ie.Changes in vitals, subtle motor movements, etc.) Were these features used in the original neurophysiologist’s interpretation?. Absolutely not, our interpretation was only based in EEG.
7. Did any patients have both findings? Often, these findings can be seen together. The reviewer is right again, because BPA and ES are sometimes observed in the same patient. However, for clarity and avoid possible biases, we preferred not use patients with both findings.